# Altered Gut Microbic Flora and Haemorrhoids: Could They Have a Possible Relationship?

**DOI:** 10.3390/jcm12062198

**Published:** 2023-03-12

**Authors:** Vincenzo Davide Palumbo, Roberta Tutino, Marianna Messina, Mauro Santarelli, Casimiro Nigro, Giacomo Lo Secco, Chiara Piceni, Elena Montanari, Gabriele Barletta, Paolina Venturelli, Girolamo Geraci, Sebastiano Bonventre, Attilio Ignazio Lo Monte

**Affiliations:** 1Euro-Mediterranean Institute of Science and Technology, 90139 Palermo, Italy; 2General Surgery 3 O.U., Molinette Hospital, University Hospital Città della Salute e della Scienza di Torino, 10126 Torino, Italy; la.tutino@gmail.com (R.T.);; 3Department of Biomedicine, Neurosciences and Advanced Diagnostics, University of Palermo, 90129 Palermo, Italy; dr.messinam@gmail.com; 4Department of Surgery, Università degli Studi di Roma “Tor Vergata”, 00173 Rome, Italy; 5Department of Surgical Sciences, University of Torino, 10126 Torino, Italy; 6Department of Surgical, Oncological and Oral Sciences, University of Palermo, 90127 Palermo, Italygirolamo.geraci@unipa.it (G.G.);

**Keywords:** microbiota, haemorrhoids, slow transit constipation

## Abstract

To date, the exact pathophysiology of haemorrhoids is poorly understood. The different philosophies on haemorrhoids aetiology may lead to different approaches of treatment. A pathogenic theory involving a correlation between altered anal canal microflora, local inflammation, and muscular dyssynergia is proposed through an extensive review of the literature. Since the middle of the twentieth century, three main theories exist: (1) the varicose vein theory, (2) the vascular hyperplasia theory, and (3) the concept of a sliding anal lining. These phenomena determine changes in the connective tissue (linked to inflammation), including loss of organization, muscular hypertrophy, fragmentation of the anal subepithelial muscle and the elastin component, and vascular changes, including abnormal venous dilatation and vascular thrombosis. Recent studies have reported a possible involvement of gut microbiota in gut motility alteration. Furthermore, dysbiosis seems to represent the leading cause of bowel mucosa inflammation in any intestinal district. The alteration of the gut microbioma in the anorectal district could be responsible for haemorrhoids and other anorectal disorders. A deeper knowledge of the gut microbiota in anorectal disorders lays the basis for unveiling the roles of these various gut microbiota components in anorectal disorder pathogenesis and being conductive to instructing future therapeutics. The therapeutic strategy of antibiotics, prebiotics, probiotics, and fecal microbiota transplantation will benefit the effective application of precision microbiome manipulation in anorectal disorders.

## 1. Introduction

The exact pathophysiology of haemorrhoids is poorly understood. Currently, the term “haemorrhoids” describes the symptomatic and abnormal downward displacement of normal anal cushions [1]; it is associated with destructive changes in the supporting connective tissue and abnormal blood circulation within anal cushions, as well as abnormal dilation and distortion of the hemorrhoidal plexus with subsequent sliding of mucosa into the anal canal. Studies on morphology and hemodynamic of the arterial supply to the anal canal have highlighted a dysregulation of the vascular tone with hyper perfusion of the hemorrhoidal plexus in patients with haemorrhoids [1,2,3]. Unfortunately, hyperperfusion does not translate into hyperoxygenation. Many studies have well demonstrated the content of inflammatory cells [4] and newly formed micro vessels [5] of hemorrhoidal tissue. Hypoxia could be considered the main responsibility of the inflammatory state characterizing haemorrhoids. Other authors proposed the theory of the internal rectal prolapse in order to explain circumferential prolapsing haemorrhoids [6]. Probably, the true pathophysiology of haemorrhoid development is multifactorial, including sliding anal cushion, hyperperfusion of haemorrhoid plexus, vascular abnormality, tissue inflammation, and internal rectal prolapse [1]. Different etiopathogenetic theories have led to different therapeutic approaches [1,7]. Haemorrhoids are strictly linked to other anorectal conditions, such as anal itching, fissures, and anismus. For most of these conditions, pathophysiology is not entirely clear, but an involvement of the smooth musculature of the internal anal sphincter has been advocated. According to current hypotheses, all anorectal disorders could be considered the direct consequence of constipation [8]. A new pathogenic theory involving a correlation between altered anal canal microflora, local inflammation, and smooth muscle impairment is proposed through an extensive review of the literature.

## 2. The Current Pathophysiology of Haemorrhoids

The cause of haemorrhoids is not clear, although, over the centuries, several theories have been postulated. Since the middle of the twentieth century, three main theories exist: (1) the varicose vein theory, (2) the vascular hyperplasia theory, and (3) the concept of a sliding anal lining. Since the time of Galen and Hippocrates, haemorrhoids were identified as varicose veins [9]. Later, this theory was disproved by Thomson and other authors who demonstrated the presence of venous dilations already in infants, suggesting that they are part of normal anal anatomy [10]. In 1956, Parks [11] attributed the varicose swellings to a local increase in pressure due to hard stools. Actually, haemorrhoids can be considered a complex system of portosystemic anastomoses [10]; in fact, haemorrhoids are no more common in patients with portal hypertension than the normal population [12]. The associated suggestion that lows fibre intake, constipation, and straining are important causative factors [13,14] is not supported by subsequent works [14,15,16,17]. Other authors, taking a cue by histologic specimens, suggested that haemorrhoids could be considered as a sort of vascular hyperplastic process [10,18] and, in 1963, Stelzner [19] proposed the concept of the corpus cavernosum recti, but this theory was confuted by Thomson [10] and Loder [18], who pointed out that there are no significant differences in the vascular anatomy between normal and pathologic hemorrhoidal tissue. Hemorrhoidal bleeding seems to arise from capillaries in the lamina propria, rather than the venous dilations [10,20]. The last and more recent theory recognizes in the pathologic slippage of the anal canal lining, the primum movens of haemorrhoids [10,18,21]. According to this theory, haemorrhoids are caused by supporting tissue disintegration or deterioration; shearing forces during defecation tend to exacerbate the problem. On microscopy, these phenomena are expressed by loss of connective fibre organization, muscular hypertrophy, fragmentation of the anal subepithelial muscle and the elastin component, and vascular changes, including abnormal venous dilatation and vascular thrombosis [10,21,22]. Ischemia due to traction and constriction of microvascular system leads to severe inflammation of the surrounding connective tissue [4]. Matrix metalloproteinase-9 (MMP-9) overexpression seems to be associated with the breakdown of elastic fibres [23] and, together with MMP-2, promotes the angioproliferative activity of transforming growth factor β (TGF-β) [24], suggesting that neovascularization might be another important phenomenon in hemorrhoidal disease. Furthermore, Chung [5] and Han et al. [23] also demonstrated that there is a higher expression of angiogenesis-related protein such as VEGF. Such growth factor contributes to the increased microvascular density in hemorrhoidal tissue. In Figure 1, a summary of current insights on haemorrhoidal disease is showed.

## 3. Constipation and Haemorrhoids

Abnormalities of anorectal physiology can be shown in patients with haemorrhoids and altered intestinal motility. Usually, one of the most common intestinal motility alterations is represented by constipation. Most patients affected by constipation have a functional disorder affecting the colon or the anorectum [25]. Constipation can be classified as (a) functional, (b) associated with irritable bowel syndrome, (c) opioid-induced, or (d) associated to functional defecation disorders, including inadequate defecatory propulsion and dyssynergic defecation. Functional constipation is characterized by prolonged delay in the transit of stool through the colon, primarily due to smooth muscle or nerve plexuses degeneration. Constipation associated to functional defecation disorders, also known as obstructive defecation [26], anismus [27], pelvic floor dyssynergia [28], or outlet obstruction [29,30], is characterized by either difficulty or inability with expelling stools from the anorectum [26]. A prolonged colonic transit and a dyssynergic defecation can co-exist. The third subtype is comprised of patients with irritable bowel syndrome and constipation (IBS-C) in whom abdominal pain, with or without bloating, is a prominent symptom, together with altered bowel habit [31]. 

Interestingly, in patients affected by hemorrhoidal disease, other intestinal dysmotility can be recognized. For instance, in patients with muscular dyssynergia, anal resting pressures is often found to be raised [16,32,33,34,35,36,37,38]. According to some authors, this abnormal finding could represent the result, rather than a cause of the pathology, suggesting a return to normal anal resting pressure within three months after haemorrhoidectomy [33,34]. Some authors have identified, in haemorrhoid hypervascularization, the possible mechanism at the base of increased anal resting pressure [36,37], but, on the other side, rubber band ligation is unable to restore normal blood pressure levels [32,33,37,38]. Recent theories have attributed dyssynergic defecation to acquired behavioural disorder or to defective learning in the childhood [25]. In these cases, a failure of recto-anal coordination due to impaired rectal contraction, paradoxical anal contraction, or inadequate anal relaxation or involuntary anal spasm (anismus), could be associated with dyssynergic defecation [26,39,40,41,42,43]. Another condition supporting the relationship between dyssynergic defecation and haemorrhoids is the demonstration of rectal ultraslow waves in patients suffering from haemorrhoids [44]. It is possible that the ultraslow waves are associated with the high resting pressures and probably originate in the internal sphincter muscle, although their significance is not clear [18]. Other changes have been recorded, but they are less reproducible, including increased external sphincter activity (spike potentials) [44], decreased anal sensation [18], and an increased number of sampling responses [44]. Unfortunately, to date, the temporal relationship between these physiologic findings and the development of haemorrhoids has not yet been explored. However, a significant slowdown of colonic propulsion mechanisms has been proven in patients with slow transit constipation [45,46]. Furthermore, it has been shown that the gastrocolic responses following a meal and the morning waking responses after sleep are also significantly diminished, but the diurnal variation of colonic motor activity is preserved [46]. In contrast, periodic rectal motor activity, a three-cycles-per-minute activity that predominately occurs in the rectum and rectosigmoid region and is invariably seen at night-time [47], significantly increases in patients with slow transit constipation [48]. This excessive uninhibited distal colonic activity may serve as a nocturnal break and retard colonic propulsion of stool [48]. Previous studies have shown that high amplitude, prolonged duration, andpropagated contractions are significantly decreased in patients affected by constipation [49,50]. Furthermore, in patients with constipation, the velocity of propagation is slower, and waves have a greater tendency to abort prematurely, and their amplitude is also decreased [49,51].

## 4. The Role of Intestinal Microorganisms in Functional Gastrointestinal Disorders

The importance of the gut microbiota in human health is currently well established. Intestinal microbiota seems to be involved in the development of the host immune system, metabolism, and digestion; it imparts specific function in the maintenance of structural integrity of the gut mucosal barrier and protection against pathogens and has a role in brain–gut communication. Tolerance to gut microbiota occurs early in life, and it is crucial to prevent allergic and immune-mediated diseases. Whenever the cross-talk between microbiota and immune system is altered, inflammation of the bowel occurs. For example, IBD (inflammatory bowel diseases) are characterized by an alteration of microbial population equilibrium, with a decrease in “good” bacteria (such as F. prausnitzii or R. hominis) and high concentration of “bad” bacteria (i.e., *E. coli*). Microbial taxa influence the immune system, hence affecting the inflammatory status of the host [52]. 

Gut bacteria are not only involved in immune system regulation, but they possibly also correlate with functional gastrointestinal disorders (FGIDs), which has been hypothesized [53]. For example, in irritable bowel syndrome (IBS), a functional disorder characterized by altered bowel transit, which is related to the use of probiotic and antibiotic therapy, has been demonstrated to bring beneficial effects to intestinal motility [54,55,56,57,58,59,60,61,62,63,64,65,66,67,68,69]. 

Another frequent and bothersome functional gastrointestinal disease is represented by chronic constipation. There is increasingly clear evidence supporting an association between the altered mucosal and faecal microbiota and chronic constipation [70,71], although precise pathophysiologic mechanisms remain poorly understood. 

In recent years, several studies have been published exploring the gut microbiome in patients with constipation. Cao and colleagues [72] reported an up-regulation of serotonin (5-HT) transporter with a subsequent reduction of 5-HT concentrations in the colonic lumen of germ-free mice that received faecal microbiota from patients with constipation. Low 5-HT levels were associated with slow intestinal transit and, furthermore, the authors highlighted changes in gut microbiota composition, with a reduction of those bacterial strains belonging to the phylum Firmicutes (Clostridium, Lactobacillus, Desulfovibrio, and Methylobacterium) and increased Bacteroidetes and Akkermansia. These findings suggest a potential role for gut microbiota in the pathogenesis of chronic constipation via increased expression of the 5-HT transporter [72]. A higher production of methane gas by metanogenic flora could be another potential mechanism able to impair intestinal muscle contractions [35,73,74,75]. 

Independently of transit time, patients with constipation show characteristic variations of microbially derived metabolites [71]. In one study, antibiotic-treated mice showed altered short-chain fatty acids (SCFA) and bile acid profiles after transfer of faecal microbiota from patients with slow-transit constipation [76]. Taxonomic profiling of the faecal microbiome from patients with functional constipation and healthy volunteers has shown decreased abundance of Bacteroides, Roseburia, and Coprococcus in the first group. Furthermore, healthy volunteers were found to have a gut microbiome enriched in genes involved in carbohydrate, fatty acid, and lipid metabolism, whereas patients with functional constipation harboured a high abundance of genes involved in methanogenic pathways, hydrogen production, and glycerol [77]. Analysis of functional gene targets in constipated and healthy females also has shown increased abundance of hydrogenogenic (hydrogen-producing) and hydrogenotrophic (hydrogen-utilizing) genes in the colonic mucosa of individuals with constipation [78]. In a cross-sectional study of eight obese children suffering from constipation and 14 obese children without constipation, functional constipation was associated with a decreased abundance of the phylum Bacteroidetes, including a significant reduction of the genus Prevotella, as well as an increased abundance of multiple genera within the phylum Firmicutes, including Blautia, Coprococcus, and Ruminococcus [79]. A recent systematic review, including seven randomized controlled trials and enrolling a total of 515 children, investigated the effects of probiotics in pediatric functional constipation. Two of the included studies, those evaluating L. reuteri DSM 17938 and B. longum [80,81], reported significantly increased defecation frequency in the treatment arm, and the meta-analysis concluded that, currently, there is insufficient evidence to support the use of probiotics for pediatric functional constipation [82]. Finally, although a low-fiber diet is a known risk factor for functional constipation in children [83], there is currently little evidence to support the use of fiber for pediatric functional constipation. Multiple systematic reviews note the sparse data and high risk of bias among the current evidence base [84,85,86,87].

Up to now, microbiota has been identified with gut bacterial flora alone, but thanks to the recent advancements, our interests in intestinal microbic flora composition have been expanded to fungi (mycobiota), viruses (virobiota), and helminthes [88]. Already in 1985, 174 patients with anal itching were submitted to perianal mycoculture. Infection by *C. albicans* was observed in all groups studied, independent of the presence of disease or anal pruritus, whereas the presence of dermatophytes was always associated with pruritus ani [89]. Actually, pruritus ani is common in those cases of stool leakage and subsequent perianal soiling, conditions that could favour local fungal flora overgrowth. Typical is the case of endurance cycling athletes who often report pain and disorders in the anal region, including inflammatory processes and functional defecation problems. Sharma [90] already underlined the role of microclimatic condition on fungal overgrowth. Subsequent inflammatory state and altered smooth muscle dysmotility with high pressure values could be a direct effect of this overgrowth. In addition, permanent microtrauma originating from constant saddle vibration leads to anal fissure and chronic inflammation, which could lead to anal pain and, as a consequence, to high sphincter pressure. The high sphincter pressure, in turn, could result in muscle hypertrophy, leading to defecation problems and diarrhoea with partial anal incontinence [91]. 

## 5. An Innovative Alternative Treatment for Anorectal Disfunction and Haemorrhoids

Currently, there are many conservative options to treat haemorrhoids and the other anorectal disorders, but, up to now, no high-quality clinical trials have shown any long-term benefit. Common local medications contain low-dose anaesthetics, corticosteroids, keratolytic, protectants, or antiseptics; the prolonged use of some of these therapies could be even detrimental and should be avoided [92]. Among conservative treatments, an important role has been played by micronized purified flavonoid fractions. These compounds may possibly improve venous tone and lymphatic outflow and may help to control local inflammation [93]. Although there is a widespread use of these supplements, their use could be justified by their mild anti-inflammatory action, but additional trials are required [92,94,95].

Up to now, faecal microbiota in ano-rectal diseases has not yet received the right attention, despite the recent insights, above cited, on its crucial role on biological intestinal homeostasis and physiological process regulation. The use of a Saccharomyces cerevisiae-based ointment for haemorrhoids has been quite diffuse in the proctologist community for several years. One study reported the beneficial effects of S. cerevisiae against Clostridium difficile infections [96]. In another study on pigs, S. cerevisiae was able to reduce the translocation of enterotoxigenic *E. coli* (ETEC), strengthening mucosal immunity [97]. Very recently, Gaziano et al. [98] proposed S. cerevisiae as a possible treatment of vulvovaginal candidiasis and bacterial vaginosis, thanks to its immunobiotic properties. The authors demonstrated that the yeast has a major role in reducing the colonization of *C. albicans* and/or G. vaginalis on human mucosal surfaces and enhancing the antimicrobial effect of standard therapeutic approaches. Furthermore, dietary supplementation of S. cerevisiae significantly decreases enteric methane production via several interrelated mechanisms, including: (i) increasing of O_2_ utilization to allow for a more favourable environment for anaerobic fermentation, which in turn leads to higher volatile fatty acids production and lower pH, which restrains the growth of methanogens and protozoa, (ii) induction of a shift in fermentation pattern toward propionogenesis, which compete with methanogens for the free H_2_ within the system, and (iii) increasing of acetogenic bacteria growth, which use H_2_ to produce acetate, which serves as another alternative sink for the free H_2_ in the system [99].

In 2017, Hager et al. [100] deepened the role of the mycobiota in the gastrointestinal tract. They found that patients with Crohn’s disease (CD) tend to have much higher levels of the fungus Candida tropicalis compared to their healthy family members, as well as two bacteria, *Escherichia coli* and *Serratia marcescens*. These three organisms work together to form robust biofilms capable of exacerbating intestinal inflammation. Candida colonization has been also highlighted in patients suffering from ulcerative colitis (UC), as well as gastric and duodenal ulcers [101]. Likely, Candida colonization seems to have a role in delaying reparative processes, while inflammation promotes its growth. These effects may create a vicious cycle in which low-level inflammation promotes fungal colonization, and fungal colonization promotes further inflammation. Both inflammatory bowel disease and gastrointestinal Candida colonization are associated with elevated levels of the pro-inflammatory cytokine IL-17. Therefore, effects on IL-17 levels may underlie the ability of Candida colonization to enhance inflammation.

In their work, Panpetch and colleagues demonstrated the correlation between *C. albicans* and inflammation in a dextran-sulfate solution (DSS) induced-colitis mouse model (DSS + Candida). Higher concentration of *C. albicans* in murine intestinal lumen seems to favour translocation of LPS, BG, and bacteria (not fungemia) from the gut into systemic circulation and causes higher mortality, more severe colon histology, and enhanced gut-leakage. The study highlighted also the presence of Pseudomonas aeruginosa in blood and faecal samples. The administration of L. rhamnosus L34 attenuated gut local inflammation, gut-leakage severity, faecal dysbiosis, and systemic inflammation. 

Other studies have reported the effects of antifungal treatment on patients affected by ulcerative colitis, observing that a reduction in fungal colonization could be beneficial for colonized patients. Data from some studies reported beneficial effects on human patients with UC after administration of Lactobacillus acidophilus [102], whereas acetic-acid treated rats who received *C. albicans* and an inhibitor of gastric acid secretion [103,104] showed reduced ulcer size. These findings suggest that, by antagonizing Candida colonization, modulation of the bacterial microbiota could provide beneficial effects for patients. 

Further studies to discern the mechanisms for the effect of inflammation on Candida colonization and the effect of Candida on inflammatory lesions represent exciting directions for future research.

From depicted data, anorectal inflammation could be the direct effect of altered bacterial and fungal intestinal flora, representing the primum movens of all anorectal disorders. Chronic constipation, dyssynergic defecation, anal fissures, and haemorrhoids could be the lowest common denominator of the same condition: dysbiosis. Microbioma composition alteration promotes inflammation and dysmotility in the whole intestinal tract, including the anal canal and rectum. Figure 2 shows a possible physiopathological hypothesis of haemorrhoids and other anorectal disorders, involving dysbiosis as the main starting point. Taking into account this new insight, an effort should be made on deepening the mechanisms that link dysbiosis to anorectal disorders in order to shift therapy from current surgical approaches to the next generation of precise mechanism-based interventions. To date, although new surgical techniques allow quicker recovery and promise to be less painful, higher recurrence rates and potentially serious complications remain an issue [105]. A deeper knowledge of the gut microbiota in anorectal disorders lays the basis for unveiling the roles of these various gut microbiota components in anorectal disorders pathogenesis, being conductive to instructing on future therapeutics. The therapeutic strategy of antibiotics, prebiotics, probiotics, and faecal microbiota transplantation will benefit the effective application of precision microbiome manipulation in anorectal disorders [106,107]. Further studies are needed in order to assess possible new non-invasive microbiota-based approaches to anorectal diseases, avoiding surgical interventions and improving patients’ satisfaction. Certainly, in the next few years, new insights on micro-/mico-/virobiota will lead to new therapeutic strategies, which are less invasive and cheaper than current surgical approaches. Local administration of “good microorganisms” against “bad microorganisms” could be a promising strategy in order to turn off inflammation and normalize internal sphincter activity, restoring a complete anal canal functionality and favouring more physiologic intestinal movements.

## Figures and Tables

**Figure 1 jcm-12-02198-f001:**
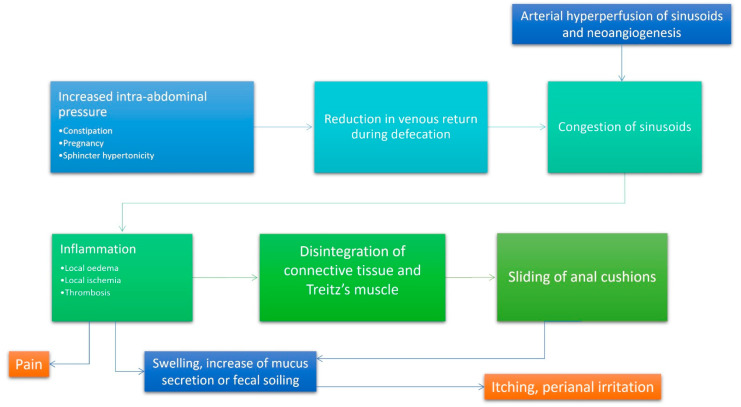
Integrated algorithm summarizing contemporary thinking regarding the pathophysiology of haemorrhoids. According to the current accepted physiopathological mechanism of hemorrhoidal disease, increased intraabdominal pressure (due to constipation, diarrhoea, pregnancy, or obstructed defecation) and arterial hyperperfusion of the hemorrhoidal plexus represents the most important initiation points. The subsequent reduction of venous return during defecation and stagnation of blood inside the dilated plexus favour inflammation and its epiphenomena (oedema, ischemia and thrombosis). Inflammation is responsible of the impaired quality of collagen and relaxation of the supporting muscles of the internal hemorrhoidal plexus. These events lead to sliding of the anal cushions that is responsible for all symptoms experienced in haemorrhoids.

**Figure 2 jcm-12-02198-f002:**
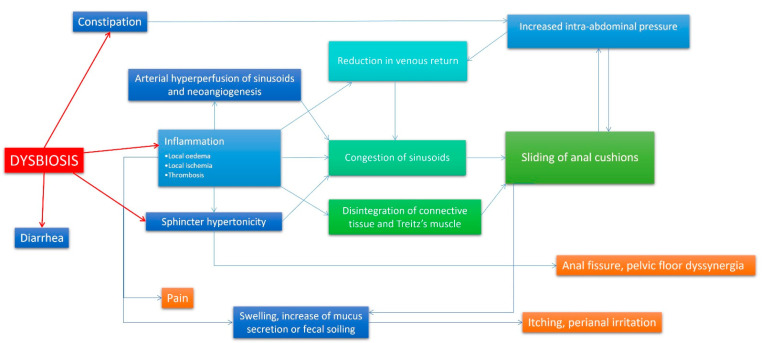
A new theory on haemorrhoidal disease pathophysiology. In this flow-chart, a new model of haemorrhoidal disease pathophysiology mechanism is presented. In this case, dysbiosis is the primum movens of the disease, and events that lead to symptoms are the direct effect of the local inflammatory state caused by altered rectal microbiota/mycobiota. vascular congestion, neoangiogenesis and arterial hyperperfusion, oedema, and connective disruption due to inflammation are responsible of swelling and cushion prolapse. Inflammation and a possible direct involvement of bacteria (producing substances able affect intestinal muscular layer) could be involved also in internal sphincter hypertonicity and subsequent increased intra-abdominal pressure. The reported mechanism is established and perpetuated with the assistance of many vicious cycles, and it is continuously auto-reinforced. In other words, haemorrhoidal disease becomes worse over time.

## Data Availability

Not applicable.

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
