# Peer review of "Altered Gut Microbic Flora and Haemorrhoids: Could They Have a Possible Relationship?"

_jcm, 2023, doi:10.3390/jcm12062198_

Round 1
Reviewer 1 Report
I read with interest the review linking microbiome concept with anorectal disorders. I find the manuscript content dominant with haemorrhoids and constipation as the two themes. This, fissure in ano, fistula in ano, pilonidal sinus, pruritus ani, etc missing in the description i.e. the title is not really reflection of the content. Similarly, the abstract and introduction are mostly of haemorrhoids and not in general about anorectal disorders. I suggest that authors please review the manuscript and revise
1. Edit the title to reflect only haemorrhoids, constipation rather than "anorectal disorders" Or else add a short paragraph/subheading of all the common anorectal conditions and link how microbiome is associated with those conditions. PMID: 34530908 gives you a clue of what anorectal problems could be included in your manuscript for microbiome discussions.
2. The constipation segment is too detailed and should have its own paragraph. Also include other pathologies as per above comment.
3. Heading/title no. 4 about intestinal microorganisms has many themes within it - one is constipation, other is about children/paediatric data, rest is all mumbo-jumbo mixture of other issues that you have grouped as muscular dyssynergia. Either go by a diagnosis of condition or go by anatomical aspect or go by clinical symptoms aspect - but please structure and organize the content. In current form, it is disorganized and haphazard in the presentation.
4. Your scope of conditions also include IBD and IBS and paediatric population. Neither IBS is not anorectal pathology. Similarly though IBD could affect anus, in general it is more of a colo-rectal pathology and not anal. So these pathologies should ideally be excluded. It appears that authors have lost focus and the description is vague and not properly organized.
5. There are many bacteria mentioned in the paper as being useful. It would be good to tabulate on name, how bacteria works/helps, etc other details - possible to tabulate good/bad bacteria or make some figure etc of those? That could be useful for readers.
Thanks
Reviewer
Author Response
Author point-by-point reply to Reviewer’s Comments
Reviewer 1
Comment #1 (C#1): Edit the title to reflect only haemorrhoids, constipation rather than "anorectal disorders" Or else add a short paragraph/subheading of all the common anorectal conditions and link how microbiome is associated with those conditions. PMID: 34530908 gives you a clue of what anorectal problems could be included in your manuscript for microbiome discussions.
Author response (AR): Agreeing with the Reviewer, the title was changed accordingly.
C#2: The constipation segment is too detailed and should have its own paragraph. Also include other pathologies as per above comment.
AR: We thank the Reviewer for this suggestion. A paragraph was dedicated to constipation.
C#3: Heading/title no. 4 about intestinal microorganisms has many themes within it - one is constipation, other is about children/paediatric data, rest is all mumbo-jumbo mixture of other issues that you have grouped as muscular dyssynergia. Either go by a diagnosis of condition or go by anatomical aspect or go by clinical symptoms aspect - but please structure and organize the content. In current form, it is disorganized and haphazard in the presentation.
AR: Actually, the content of this paragraph could appear confuse, but we think that the role of microbiota on intestinal motility was depicted in a satisfying manner, at least in its most important aspects. In order to avoid confusion and address your request, the title was changed.
C#4: Your scope of conditions also include IBD and IBS and paediatric population. Neither IBS is not anorectal pathology. Similarly though IBD could affect anus, in general it is more of a colorectal pathology and not anal. So these pathologies should ideally be excluded. It appears that authors have lost focus and the description is vague and not properly organized.
AR: Once the title has been changed, we think the paragraph will be more readable and comprehensible. IBD and IBS are only example how microflora can affect intestinal immune system and intestinal motility. These two aspects are involved in haemorrhoids and other anorectal conditions (as shown in figure 1 and 2). The main theme of the paragraph are microorganisms and their actions on intestinal wall (from mouth to anus), therefore we cannot address the request expresses in this comment.
C#5: There are many bacteria mentioned in the paper as being useful. It would be good to tabulate on name, how bacteria works/helps, etc other details - possible to tabulate good/bad bacteria or make some figure etc of those? That could be useful for readers.
AR: A table was added (Table 1).

Reviewer 2 Report
I would like to congratulate the authors for the manuscript. There is a lot of writing, and the topic is of great relevance and very interesting. During the text, the authors also raised several problems to be discussed regarding the theme.
However, I would like to ask a few questions about the topic.
Between lines 178 to 181 the authors speak of an association with serotonin. I would like to know if you have investigated any studies that might associate some mental or psychic disorder, such as depression, with cases of hemorrhoids?
Between lines 196 to 199 the authors speak of an association between some genes. I would like to know what these genes are? I think it would be important to unpack this a little bit. These could be important markers, in addition, their regulation could be therapeutic targets. My suggestion is that the authors group them in a table and that it contains their functions as well.
During my reading, I would suggest a topic only on probiotics and prebiotics and their association with hemorrhoids, however, this subject was very well explained in topic 5. I also suggest creating a topic, or ending topic 5, addressing and suggesting hypotheses in which direction studies with treatments for hemorrhoids will go.
Author Response
Author point-by-point reply to Reviewer’s Comments
Reviewer 2
C#1: Between lines 178 to 181 the authors speak of an association with serotonin. I would like to know if you have investigated any studies that might associate some mental or psychic disorder, such as depression, with cases of haemorrhoids?
AR: About the first comment, actually we have never explored the association between some mental or psychic disorder, such as depression, and haemorrhoids. Undoubtedly, there is a strong association between intestinal bowel disease and anorectal disorders (haemorrhoids and anal fissures). IBS is, in turn, associated often with anxiety and other mental illnesses. Interestingly, a work by Sit M et al (Sit M, Yilmaz EE, Canan F, Yıldırım O, Cetin MM. The impact of type D personality on health-related quality of life in patients with symptomatic haemorrhoids. Prz Gastroenterol. 2014;9(4):242-8.) assessed the prevalence of type D personality in patients with haemorrhoids. Considering that, currently, there no many studies analysing this correlation, it could be object of possible researches in the future.
C#2: Between lines 196 to 199 the authors speak of an association between some genes. I would like to know what these genes are? I think it would be important to unpack this a little bit. These could be important markers, in addition, their regulation could be therapeutic targets. My suggestion is that the authors group them in a table and that it contains their functions as well.
AR: About the second question, the articles cited in the text (references 76 and 77) already explain clearly the differences in genetic expression of bacteria involved in constipation and other intestinal motility alterations. Attached you can find a table (Table 2) with a list of pathways detected in these analyses, but we fear it may prove redundant for our discussion. Nevertheless, if you want, we can add it into the text.
C#3: During my reading, I would suggest a topic only on probiotics and prebiotics and their association with hemorrhoids, however, this subject was very well explained in topic 5. I also suggest creating a topic, or ending topic 5, addressing and suggesting hypotheses in which direction studies with treatments for hemorrhoids will go.
AR: A possible answer to the last comment, is highlighted at the end of the article.

Round 2
Reviewer 1 Report
Dear Authors
Thanks for the minor selected edits that you have made. I see that many suggestions are deemed as unnecessary and ignored. Some of such issues are partly acceptable. For example, I agree that you can retain IBD and IBS to allude to dysbiosis associated with these conditions. However, to entirely disregard the issue about including the paediatric population is, in my opinion, unacceptable.
1. I suggest that the section on paediatric population should be omitted.
2. Figure 2 vector is wrong to suggest that anal cushions raise abdominal pressure, it should be the other way round (the direction of arrow).
3. I somehow am intrigued by the saddle vibration and microtrauma as pathology for anal fissures (page 6 line 237). There is no citation/reference given.
4. Lastly, though authors disagree with my earlier comments, i still feel there is a lot of anorectal muscular and sphincter pathophysiology + constipation pathophysiology which is quite out of the scope of dysbiosis actually. There is too much emphasis on those pathophysiology issues than is warranted, considering main scope is dysbiosis and not pathophysiology of how the muscle works etc.
5. Though authors have added the Table, i am unable to see the Table and may be this point is not for authors but for editorial office to check and upload etc.
Thanks
Author Response
Author point-by-point reply to Reviewer’s Comments
Reviewer 1
Comment #1 (C#1): I suggest that the section on paediatric population should be omitted.
Author response (AR): We thank the reviewer for the constructive comments. Although in the full respect of your suggestion, we would propose that the studies on pediatric population should be maintained because microbiota composition in adults is strictly related to childhood. As depicted in our article, dysbiosis in paediatric population exerts negative effects on intestinal motility and explain the establishment of functional gastrointestinal disorders. This aspect is highlighted also by other studies:
-Srinivasjois R, Gebremedhin A, Silva D, Rao S, Pereira G. Probiotic supplementation in neonates and long-term gut colonisation: A systematic review of randomised controlled trials. J Paediatr Child Health. 2023 Feb;59(2):212-217
- Korpela K, de Vos WM. Infant gut microbiota restoration: state of the art. Gut Microbes. 2022 Jan-Dec;14(1):2118811
C#2: Figure 2 vector is wrong to suggest that anal cushions raise abdominal pressure, it should be the other way round (the direction of arrow).
AR: We agree with this reviewer and according to the suggestion we propose to insert a double vector as abdominal pressure pushes the anal cushions but further according to our theory, low rectum inflammation and oedema with subsequent prolapse of anal cushions could contribute to obstructed defecation and consequent abdominal pressure rise (ref. 35-36).
C#3: I somehow am intrigued by the saddle vibration and microtrauma as pathology for anal fissures (page 6 line 237). There is no citation/reference given.
AR: We thank the reviewer for the comment, we included ref. n. 90 that focuses on this aspect.
C#4: Lastly, though authors disagree with my earlier comments, i still feel there is a lot of anorectal muscular and sphincter pathophysiology + constipation pathophysiology which is quite out of the scope of
dysbiosis actually. There is too much emphasis on those pathophysiology issues than is warranted, considering main scope is dysbiosis and not pathophysiology of how the muscle works etc.
AR: We thank the reviewer for his observation. In our work we would underline the relationship between intestinal microorganisms, inflammation and intestinal motility. In the paragraph dedicated to constipation and other functional disorders, we have highlighted those alterations that can influence the activity of the muscular layer. Taking into account the several studies demonstrating the effects of intestinal microorganisms on the muscular layer (ref. 71-86), we hypothesized the possible role of microorganisms on intestinal motility (including internal sphincter activity).
C#5: Though authors have added the Table, i am unable to see the Table and may be this point is not for authors but for editorial office to check and upload etc.
AR: We hope this problem will be solved by the editorial office.
Reviewer 2 Report
I have no further considerations or suggestions. The authors significantly improved the manuscript. In my opinion, the manuscript should be published as an article in the format it is.
Author Response
Thank you for your comments